# The research on the effect of temperature of electro-surgical unit to surgical smoke distribution in theatre-in vitro and simulation study

**Hui Yu** *

Department of Gynaecology and Obstetrics, Huazhong University of Science and Technology Hospital, Wuhan, Hubei Province, China

* jinyan203558@126.com

**Citation:** Yu H (2024) The research on the effect of temperature of electro-surgical unit to surgical smoke distribution in theatre-in vitro and simulation study. PLoS ONE 19(3): e0299369. https://doi.org/10.1371/journal.pone.0299369

**Data Availability Statement:** All relevant data are within the manuscript and its Supporting information files, or have been deposited at the following figshare page: (DOI: 10.6084/m9.figshare.24936747).

## Abstract

In electro-surgery, surgical smoke was hazard to surgeons and patient in theatre. In order to institute effective countermeasures, quantifying of the effect of tip temperature of electro-surgical unit to surgical smoke distribution in theatre was studied. The relation of tip temperature to power of electro-surgical unit through in vitro cutting experiment. Based on experiment data, the mathematical model was established to simulate the electro-surgery in laminar operation room. As the power of electro-surgical knife increased, the knife tip temperature increased. Total content of ($CO$, $CO_2$, $CH_4$, $NH_3$) in waste gas and net flow rate of waste gas at outlet increased with the rising temperature of knife tip and formation rate of condensed tar droplets and non-viable particles also increased. Based on simulation, it was found that The maximum height of surgical smoke rising right above the incision of electro-surgical unit was increased with rising temperature of electro-surgical knife tip. There was a spread route of dispersed surgical smoke near the walls of theatre through natural convection. The polynomial fitting relationship was derived. As the tip temperature of knife increased from 200 to 500°C, maximum ascending height of surgical smoke right above the incision position of electro-surgical unit increased from 1.1 m to 1.45 m. When the tip temperature of electro-knife was more 400°C, the CO content in the surgeon's operating zone was more than 200 ppm, which would cause the surgeon's HbCO level increased. As the patient's tissue in the wound during operation was open, when the electro-knife of more than 400°C, the content of condensed tar droplets and in-viable particle was higher than 20 g/m³ and 12 g/m³ in the zone around patient's wound of open tissue, which should be hazard to health of patient.

## Introduction

In electro-surgery, surgical smoke was a health risk to person in theatre. The surgical smoke had a pungent odor, hindered the surgeons' vision, and caused irritation of eyes. Many studies show that it contained harmful substances [1–4].

**Funding:** The author(s) received no specific funding for this work.

**Competing interests:** NO authors have competing interests.

Surgical smoke was sourced from ablation of tissue protein and fat. It was observed that surgical smoke was smoke or plume produced by cautery or diathermy, and there was aerosols filled in theatre [5, 6]. The air stream, thermal of electro-surgical knife and volatizing of tissues drove the smoke to spread in the whole theatre [7–9].

The potential hazards of surgical smoke were carbon monoxide, toxic chemicals and non-viable particles [10–12]. The production of CO was common during tissue ablation, and the CO level was abundantly increased after an electro-surgery [13]. The non-viable particles produced in electro-surgery were ultrafine with size of 0.5–5μm, and traveled long in theatre [14]. It was found that electro-surgical coagulation induced the production of a high number concentration ($> 100,000 \text{ cm}^{-3}$) of particles. The peak concentration was confined to the near surrounding of the surgical bed [15]. There were many kinds of toxic chemical compounds in surgical smoke, and the most toxic chemicals were produced in electro-surgery in the formed of volatized tar [16]. The common toxic chemical compounds were polyaromatic hydrocarbons, such as benzene, nitriles and phenols [17–20].

As the power level of electro-surgical unit increased, tip temperature of knife rose, and production of surgical smoke increased [21]. And the flow field and concentration field in theatre changed with the raising of tip temperature of knife. In order to effective protection of person in theatre during electro-surgery, it was necessary to understand the effect of tip temperature of knife on surgical smoke distribution in threatre during with simulation of mathematical model.

This paper focused on the effect of tip temperature of knife on distribution of waste gas, tar and non-viable particles in theatre, and based on the results of the model estimation, the risk of the surgeon and patient after electro-surgery was analyzed.

## Methods

### In vitro cutting tissue operation

The tissue cutting experiments of back muscle tissue in miniature porcine with the electro-surgical knife at certain power levels were carried out for in vitro study of the effect of power level of electro-surgical knife on quantity of waste gas, non-viable particles and tar generated from cutting position. Based on the experiment data, the mathematical model was established to simulate the flow and concentration field in theatre during electro-surgery with the aim to study the effect of tip temperature of knife on distribution of waste gas, tar and non-viable particles in theatre.

Muscle tissue samples were collected in back section of a dead 90-day-old miniature porcine bought in market, and then were sliced into 10×5×5cm pieces, which were put in stainless container and then placed on ice. In the experiment, the tissue piece was fixed on a plate in a stainless crucible with sealing lid, the plate could move horizontally by drawing of shaft outside and the shaft was sealed with rubber through the wall of crucible, as shown in Fig 1. The electro-surgical unit (DGD-300B-2) could insert through the rubber sealing hole of lid into crucible and reached 5mm cutting depth of sample tissue piece. Air flowed $4{\times}10^{-5}$ m$^3$/s into stainless crucible through the stainless inlet pipe on lid of crucible and flow out through outlet pipe on the lid. There were a water-cool tube, a filter section and rotor flowmeter connected in the stainless outlet pipe. Tar formed in crucible during experiment was collected in water-cool tube, non-viable particles were collected in the filter section, and the outflow rate of air was measured with LZB6 rotor flowmeter.

During experiment, the muscle tissue sample piece was put on the plate in crucible, lid of crucible was covered and sealed, air flowed into the crucible, electro-surgical knife was inserted through the rubber sealing hole on the lid and turned on, the power of the electro-surgical

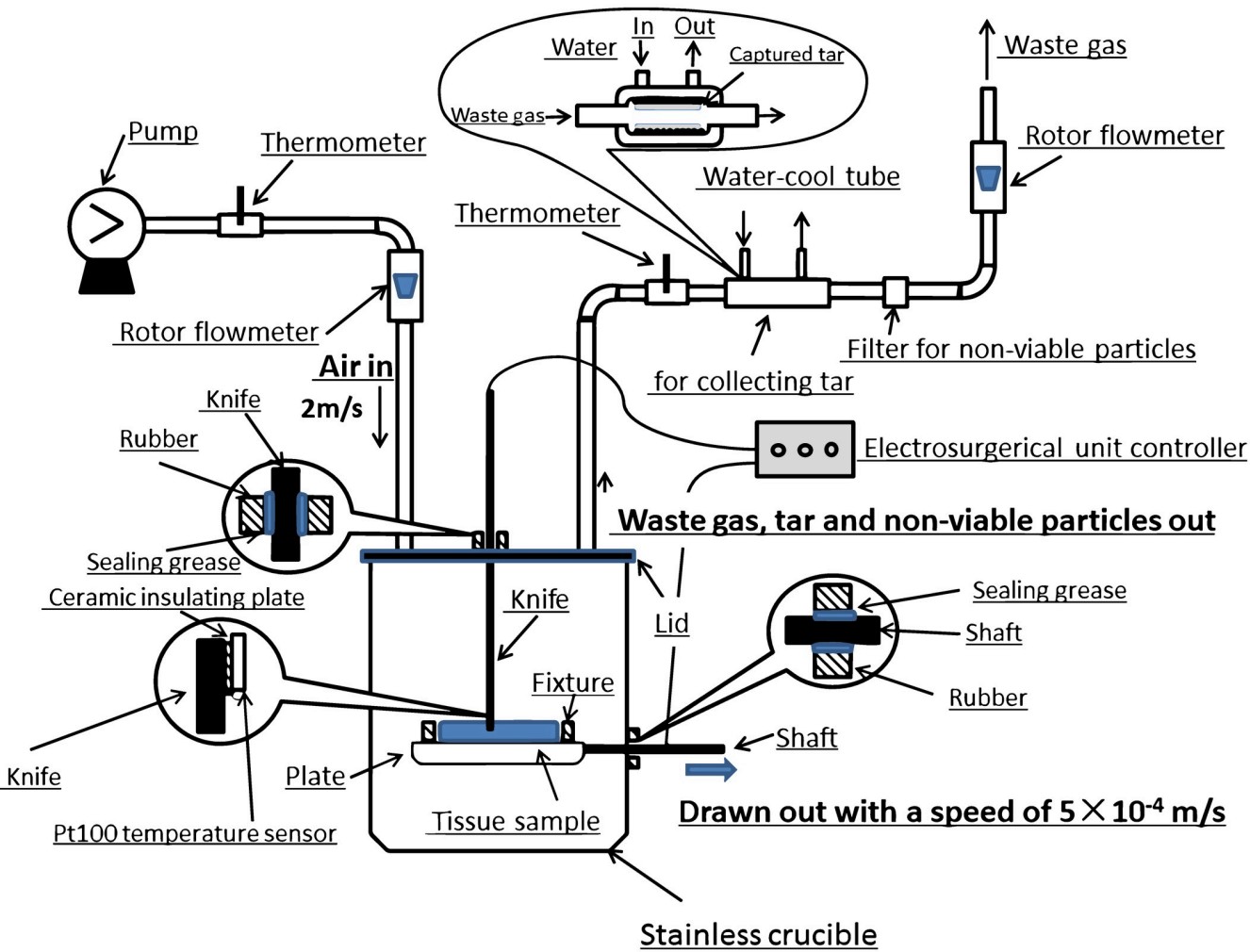

**Fig 1. Schematic diagram of in vitro cutting tissue operation.**

knife was lifted up to certain level(20, 30,40, 50, 60, 70 and 80W), then the electro-surgical unit was inserted into tissue sample with depth of 5 mm, then the sample fixed on the plate was drawn horizontally with a speed of $5\times10^{-4}$ m/s by the shaft driven outside, and the sample was cut by the electro-surgical unit by this method. The waste gas flowed out the crucible with tar and non-viable particles through the outlet stainless pipe, and was cooled in a water-cooled tube to collect the tar, and then passed filter to collect non-viable particles inside the outflow gas from crucible. After each experiment, water-cool tube was washed with $CH_2Cl_2$ to dissolve and tar released as tissue pyrolysis was collected, and then $CH_2Cl_2$ was evaporated on a water bath, and finally tar was weighted. The physical property of tar was measured with DAM4500 densimeter and SYD-265C-1 kinematic viscosimeter. The non-viable particles were captured with filter section in out-stream pipe, and the weight of non-viable particles was measured with electronic balance. Finally, the flow rate of waste gas was gauged with rotor flowmeter, and temperature of waste gas was measured with Pt100 temperature sensor, and then the flow rate of waste gas at STP (standard temperature and pressure) was converted with consideration of the difference between mass flow rate of inflow and outflow. There was Pt100 temperature

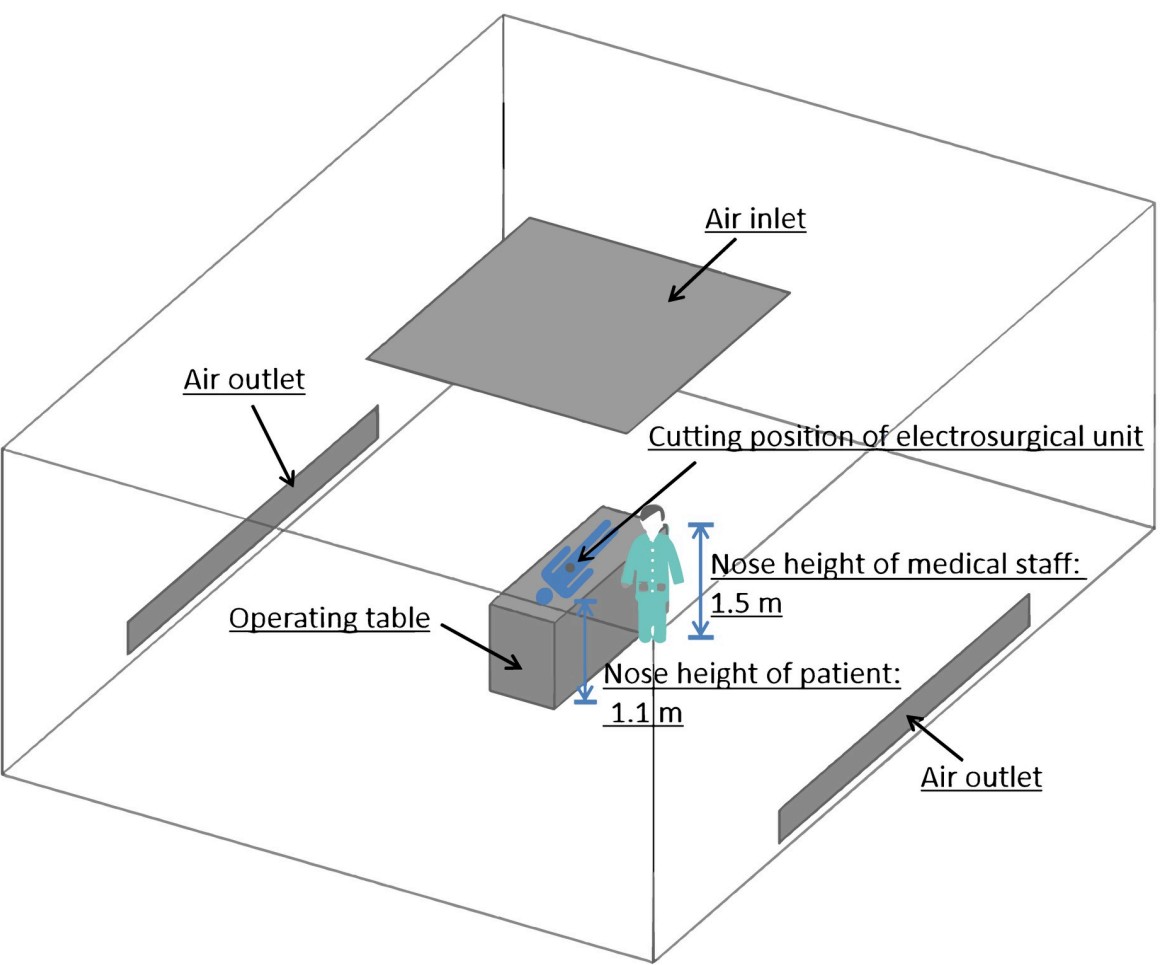

**Fig 2. A laminar flow operation room during electro-surgery, which was simulated with mathematical model.**

sensor strapped to the electro-surgical unit with the distance of 5mm to the tip of the knife with the aim to measure the gas temperature at the incision of the electro-surgical knife and there was a thin ceramic insulating pad between the sensor and knife to keep out of the influence of electrical current on the knife. As the gas at the incision of the electro-surgical unit was sourced by the nearby knife tip, the gas temperature was set as the tip temperature of the electro-surgical knife. The composition of outflow gas from stainless crucible was measured with ZGC-2100 gas chromatography.

## Mathematical model

Based on the quantity and property data of waste gas, tar and non-viable particles measured in the above in vitro cutting tissue operation, a mathematical model was established to simulate the flow field in theatre during electro-surgery with the aim to understand the distribution of non-viable particles or tar droplets in the theatre.

The flow field of a laminar flow operation room (Fig 2) during laparotomy with electro-surgical knife was simulated with mathematical model. The theatre was 8 m long, 6 m wide and 3 m high. The fresh air flowed vertically down from an inlet (2.4 m×2.6 m) on the ceiling, and

the waste gas in theatre flowed out through two outlets (4 m×0.3 m, the lower edge of outlet was 0.1 m off the floor) on the two opposite walls in theatre. There was an operating table (1.8 m×0.8 m×0.8 m) in the theatre. The cutting position of electro-surgical unit in a patient on the operation bed was assumed at the point 0.1m above the center of the operating table.

As a complex process, the flow behavior of gas and the temperature change in theatre during electro-surgery was simulated with the equation of continuity equation, the momentum equation of Navier-Stokes equation, describing the turbulence model of $k$-$\varepsilon$ two equation model and energy equation. The waste gas formed from the tip of the electro-surgical knife was simulated with the species transport equation. The tar and non-viable particles formed at the cutting position were tracked with the discrete phase model.

(1) The continuity equation:

$$\frac{\partial \rho}{\partial t} + \frac{\partial \rho u_i}{\partial x_i} = 0 \tag{1}$$

In the formula, $u_i$ was the time averaged velocity expressed by tensors, m/s; $\rho$ was pressure (Pa); $t$ was time(s);

(2) The Navier-Stokes equation:

$$\frac{\partial \rho u_i}{\partial t} + \frac{\partial \rho u_i u_j}{\partial x_j} = \frac{\partial p}{\partial x_i} + \frac{\partial}{\partial x_j}\left[\mu_{\text{eff}}\left(\frac{\partial u_i}{\partial x_j} + \frac{\partial u_j}{\partial x_i}\right)\right] + \rho g_i \tag{2}$$

Where $u_i$ and $u_j$ denoted the velocity of i and j direction, m/s; $x_i$ and $x_j$ stood for coordinates i and j direction, m; $\rho$ and $p$ was fluid density(kg/m$^3$) and pressure(Pa); $u_{\text{eff}}$ indicated the coefficient of effective viscosity(Pa·s) determined by the $k$-$\varepsilon$ two equation turbulence model;

(3)The $k$-$\varepsilon$ two equation model
k equation of turbulent kinetic energy:

$$\frac{\partial \rho k}{\partial t} + \frac{\partial}{\partial x_i}\left[\rho u_i k - \frac{\mu_{\text{eff}}}{\sigma_k}\frac{\partial k}{\partial x_i}\right] = G_k - \rho\varepsilon \tag{3}$$

In the formula, $k$ represented turbulent kinetic energy, m$^2$/s$^2$, and $\varepsilon$ denoted the dissipation rate of turbulent flow energy, m$^2$/s$^3$;

$\varepsilon$ equation of dissipation rate of turbulent energy:

$$\frac{\partial \rho\varepsilon}{\partial t} + \frac{\partial}{\partial x_i}\left[\rho u_i\varepsilon - \frac{\mu_{\text{eff}}}{\sigma_\varepsilon}\frac{\partial \varepsilon}{\partial x_i}\right] = (C_1\varepsilon G_k - C_2\rho\varepsilon^2)/k \tag{4}$$

Where the $G_k$ and $u_{\text{eff}}$ could be calculated by equations as follows:

$$G_k = \mu_\varepsilon \frac{\partial u_j}{\partial x_i}\left[\frac{\partial u_i}{\partial x_j} + \frac{\partial u_j}{\partial x_i}\right] \tag{5}$$

$$\mu_{\text{eff}} = \mu + \mu_t = \mu + \rho C_\mu k^2/\varepsilon \tag{6}$$

In the formula, $\mu_t$ represented the turbulent viscosity coefficient, Pa·s, and $\mu$ was the viscosity coefficient, Pa·s; $\mu_i$ (i = 1,2,3) denoted the velocities of the three axes X, Y, and Z respectively; $C_1$, $C_2$, $C_\mu$, $\sigma_k$, $\sigma_\tau$ were empirical constant, $C_1 = 1.43$, $C_2 = 1.93$, $C_\mu = 0.09$, $\sigma_k = 1.0$, $\sigma_\varepsilon = 1.43$.

(4) The energy equation:

$$\frac{\partial \rho u_i H}{\partial x_i} = \frac{\partial}{\partial x_i}\left[k_{\text{eff}}\frac{\partial T}{\partial x_i}\right] \tag{7}$$

Where $H$ was enthalpy of gas in theatre, $T$ denoted gas temperature, and $k_{\text{eff}}$ was effective heat transfer coefficient, which could be calculated as follow:

$$k_{\text{eff}}\frac{\partial T}{\partial x_i} = \frac{\mu}{\sigma_T} + \frac{\mu_t}{\sigma_{t,T}} \tag{8}$$

In the formula, $\sigma_T$ and $\sigma_{t,T}$ were empirical constant, $\sigma_T = 1.00$; $\sigma_{t,T} = 0.9$.

(5) The species transport in the gas flow is described with the transport equation.

$$\frac{\partial \rho Y_i}{\partial t} + \frac{\partial \rho Y_i u_j}{\partial x_j} = \frac{\partial}{\partial x_j}\left[\tau_i \frac{\partial Y_i}{\partial x_j}\right] + R_i + S_i \tag{9}$$

Where $Y_i$ is the mass ratio of species i, $\tau_i$ is the mass transfer coefficient of species i, $R_i$ is the generation/consumption rate of species i, $S_i$ is the source term. In this paper $R_i = 0$ and $S_i = 0$.

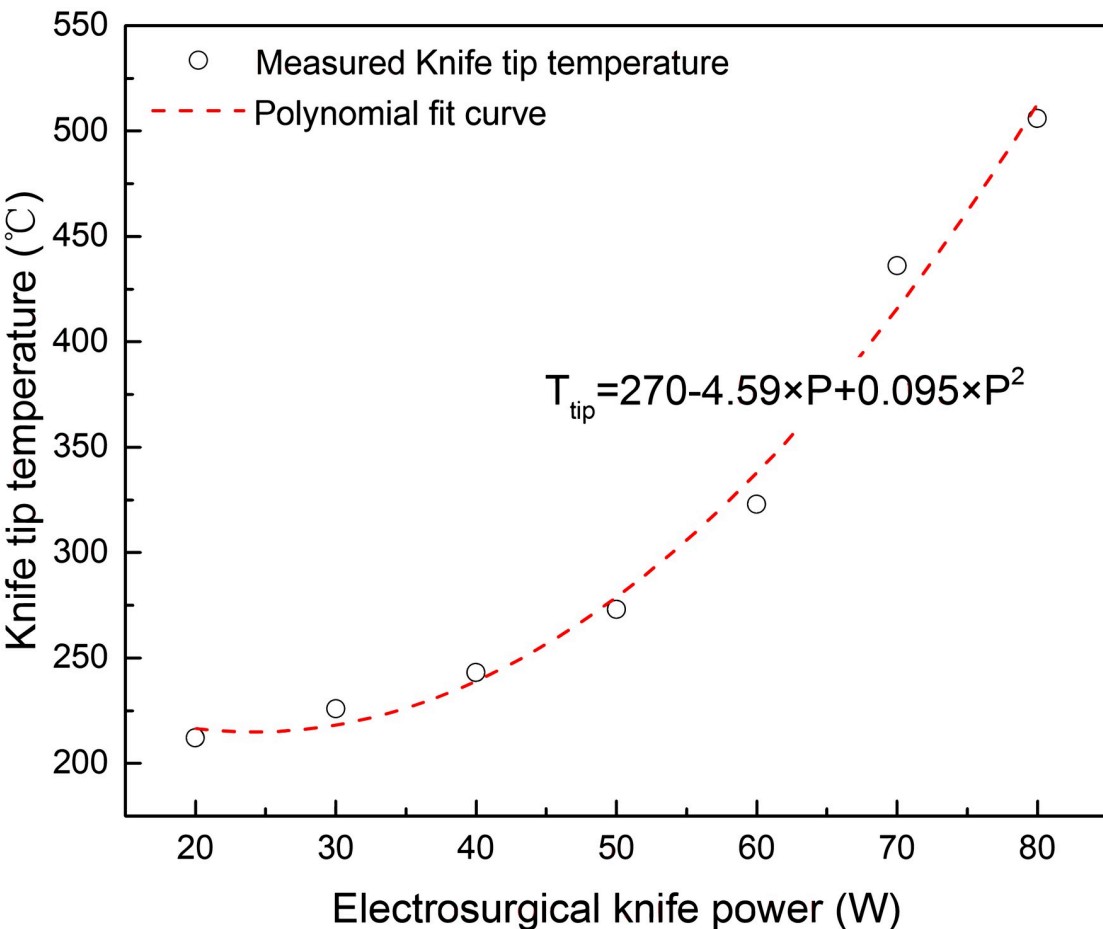

**Fig 3. Electro-knife tip temperature at different power levels in vitro experiment.**

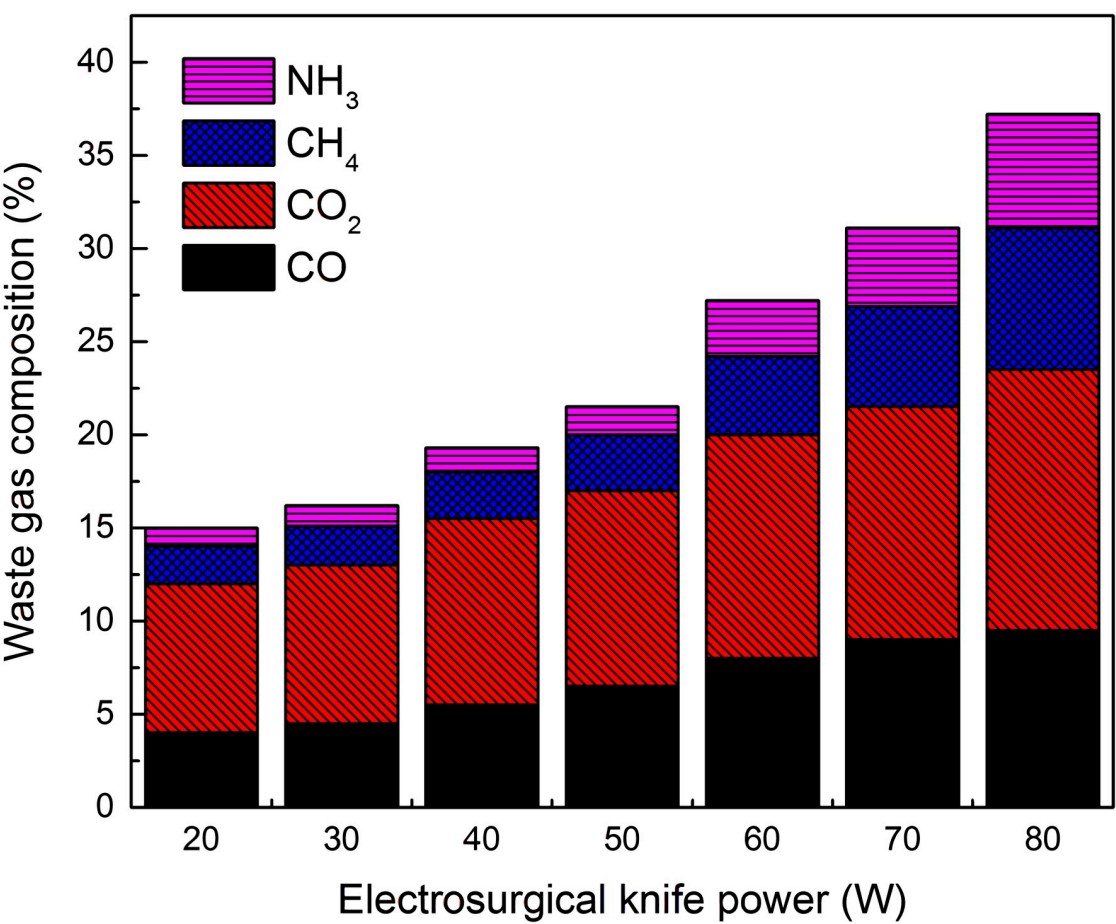

**Fig 4. Composition in waste gas at different power levels of electro-knife in vitro experiment.**

(6) The trajectory equation of non-viable particles and tar droplets

$$\frac{\mathrm{d}u_{p,i}}{\mathrm{d}t} = \frac{18\rho\mu}{\rho_p d_p^2}\left(1 + 0.15\mathrm{Re}_p^{0.687}\right)\left(u_i - u_{p,i}\right) \tag{10}$$

Where $u_p$ was velocity of non-viable particles or tar droplets, m/s; $\rho_p$ was density of non-viable particles or condensed tar droplets, kg/m³; $d_p$ was diameter of non-viable particles or tar droplets, m; $\mathrm{Re}_p$ was Reynolds number of non-viable particles or tar droplets.

Boundary conditions of the model were summarized as follows:

1. The laminar air entrance was defined at the inlet of in the ceiling of the theatre, entrance velocity ($V_{in}$ = 0.096 m/s) determined by air down flow rate in ambient steady state (27˚C); The turbulent kinetic energy($k_{inlet}$) and turbulent energy dissipation rate($\varepsilon_{inlet}$) at the entrance are calculated by Eqs (11) and (12) respectively, in which the turbulence intensity $T_i$ was 3.7%, and the turbulence length scale $l$ was 0.07 times of inlet width, and Cµ was

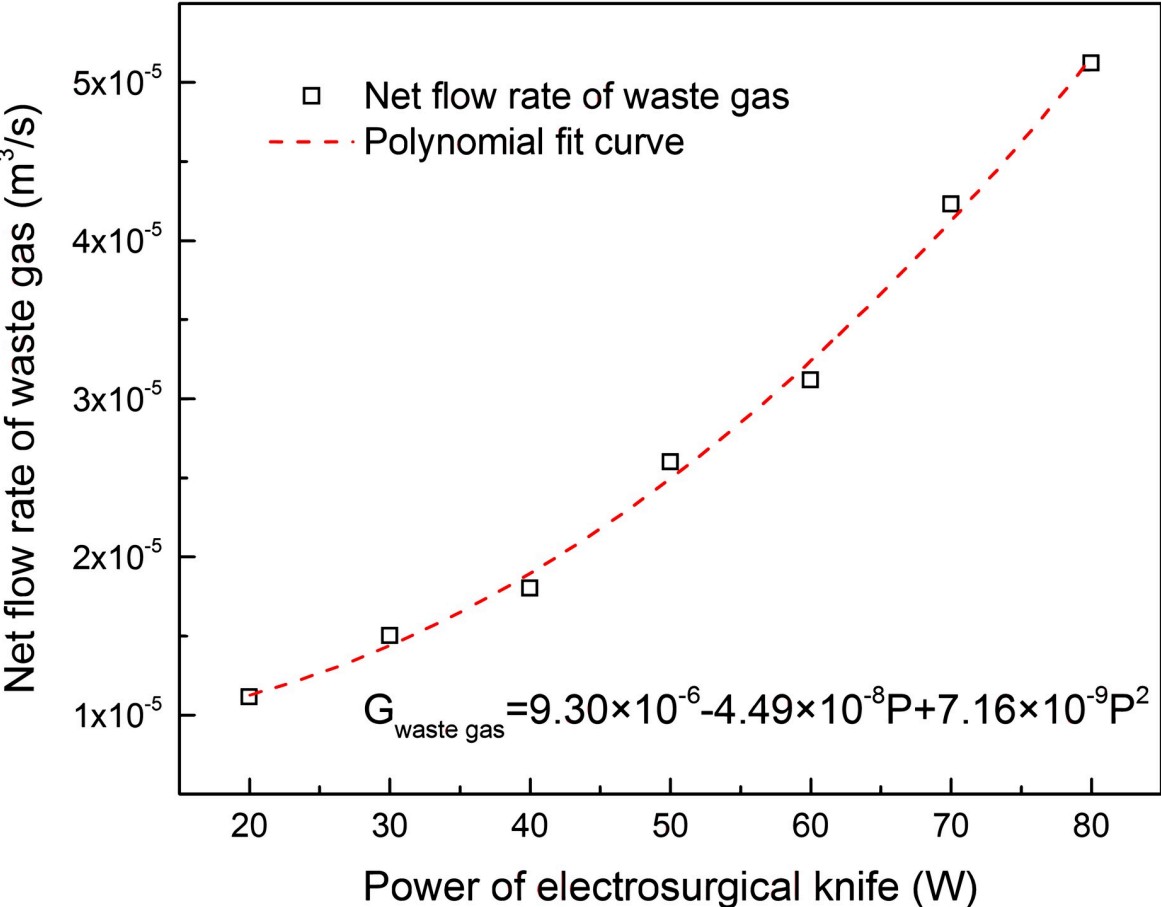

**Fig 5. Net flow rate of waste gas at different power levels of electro-knife in vitro experiment.**

taken as 0.09;

$$k_{\mathrm{inlet}} = 1.5\left(V_{\mathrm{in}}T_{\mathrm{i}}\right)^{2} \tag{11}$$

$$\varepsilon_{\mathrm{inlet}} = \mathrm{C}_{\mu}^{0.75}k^{1.5}/l \tag{12}$$

2. The air exit was set up at the bottom of wall as 'pressure outlet';

3. The cutting position of electro-surgical unit was set as an inlet of electro-surgical smoke, which included waste gas species, non-viable particles and condensed tar droplets. The flow rate, property and temperature of waste gas, non-viable particles and condensed tar droplets, were based on measurement data in the above mentioned experiment;

4. The wall of the theatre with no slip surface was treated by standard wall function, and the heat transfer at the walls was third boundary conditions with heat transfer coefficient of 1.5 and outer temperature set as 30°C;

5. The ceiling and floor of the theatre was treated as no slip surface and adiabatic boundary.

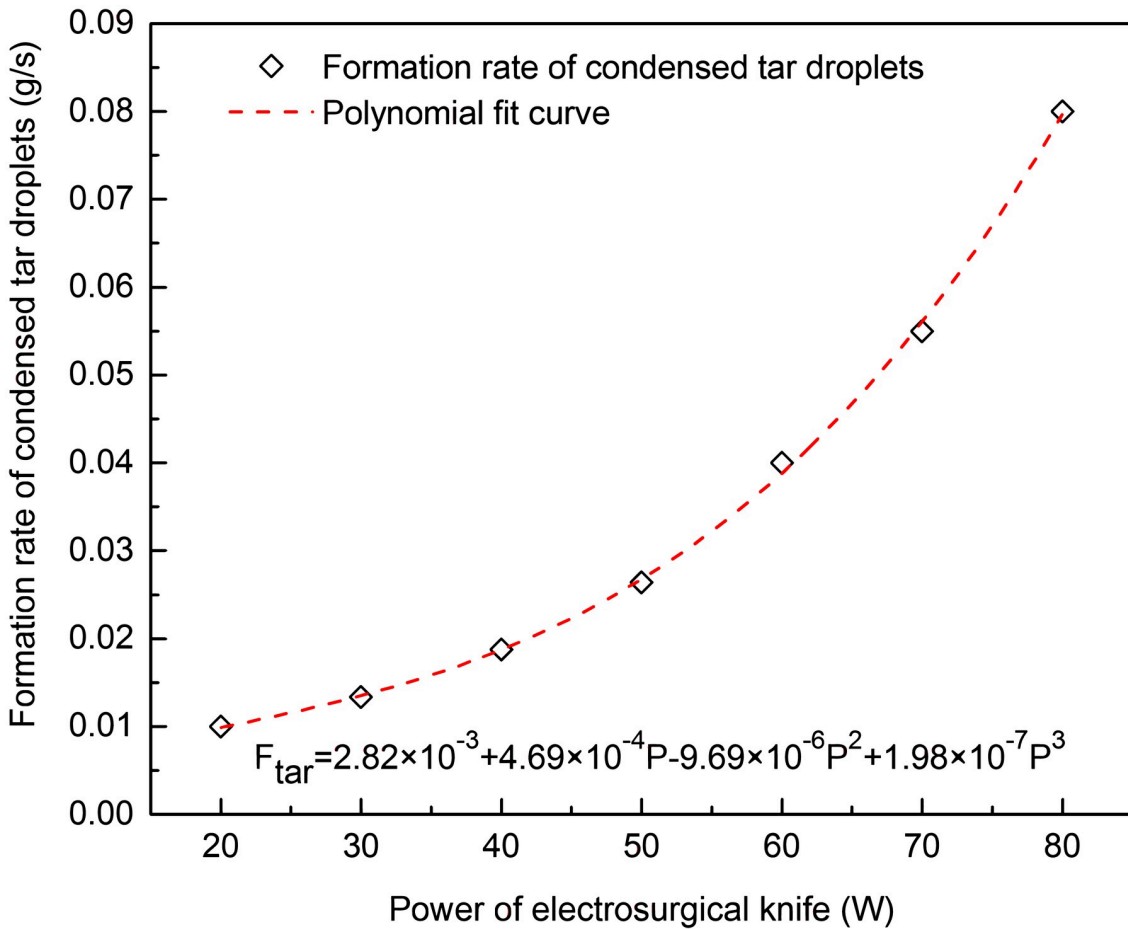

**Fig 6. Formation rate of condensed tar droplets at different power levels of electro-knife in vitro experiment.**

The mathematical model was computed in FLUENT software and the grid division($3.4 \times 10^5$ nodes created) was carried out with ICEM software. For the distribution of condensed tar droplets and non-viable particles, the multiphase mixture model was used, and the diameter of condensed tar droplets and non-viable particles were assumed as 2μm, according to the other researchers' work [17–20]. SIMPLEC algorithm (Semi-Implicit Method -Consistent) was used in the model to simulate the three-dimensional flow field and temperature field in the theatre, equation residual value was less than $10^{-4}$ as the convergence criteria solving the model equations. The thermophysical properties of materials in theatre were shown in S1 Table.

## Results

### Results of in vitro experiment

The knife tip temperature, composition and volume flow rate was measured in vitro cutting experiments of porcine back muscle with electro-surgical unit under different power levels, as shown in Figs 3–5. The formation rate of tar and non-viable particles was also measured in the experiments, Figs 6 and 7. The properties of waste gas, tar and non-viable particles were measured and listed in S1 Table (tar would transfer to condensed droplet when tar in gas phase released from the cutting position and temperature dropped to ambient.

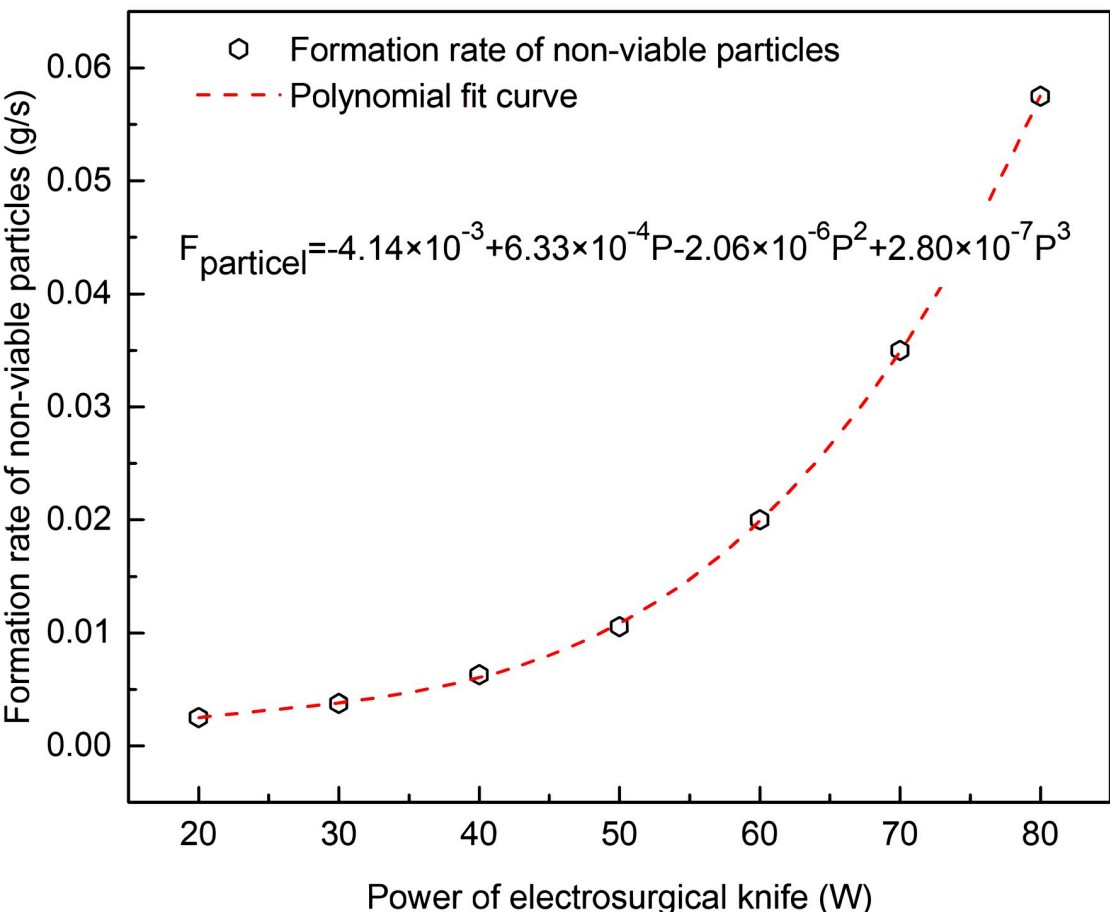

**Fig 7. Formation rate of non-viable particles at different power levels of electro-knife in vitro experiment.**

From Figs 3–5, it was found that as the power of electro-surgical knife increased, the knife tip temperature, total content of ($CO$, $CO_2$, $CH_4$, $NH_3$) in outlet waste gas and net flow rate of waste gas at outlet increased. Net flow rate of waste gas was the difference between inlet and outlet flow rate of experiment facility during in vitro cutting experiment with electro-surgical unit. At outlet, composition of ($CO$, $CO_2$, $CH_4$, $NH_3$) was sourced from the cutting position during experiment and content of $CO+CO_2$ was much higher than content of $CH_4+NH_3$ in waste gas. So the main composition of waste gas at cutting position was $CO+CO_2$. Eqs (13) and (14) described the effect of power level of electro-surgical knife to the knife tip temperature and net flow rate of waste gas at outlet.

$$T_{\text{tip}} = 270 - 4.59 \times P + 0.095 \times P^2 \tag{13}$$

$$G_{\text{waste gas}} = 9.30 \times 10^{-6} - 4.49 \times 10^{-8}P + 7.16 \times 10^{-9}P^2 \tag{14}$$

Where, $T_{\text{tip}}$ was the knife tip temperature, ˚C; $P$ was the power of electro-surgical knife, W; $G_{\text{waste gas}}$ was net flow rate of waste gas at outlet, i.e. the difference between inlet and outlet flow rate of experiment facility during in vitro cutting experiment with electro-surgical unit, $m^3$/s.

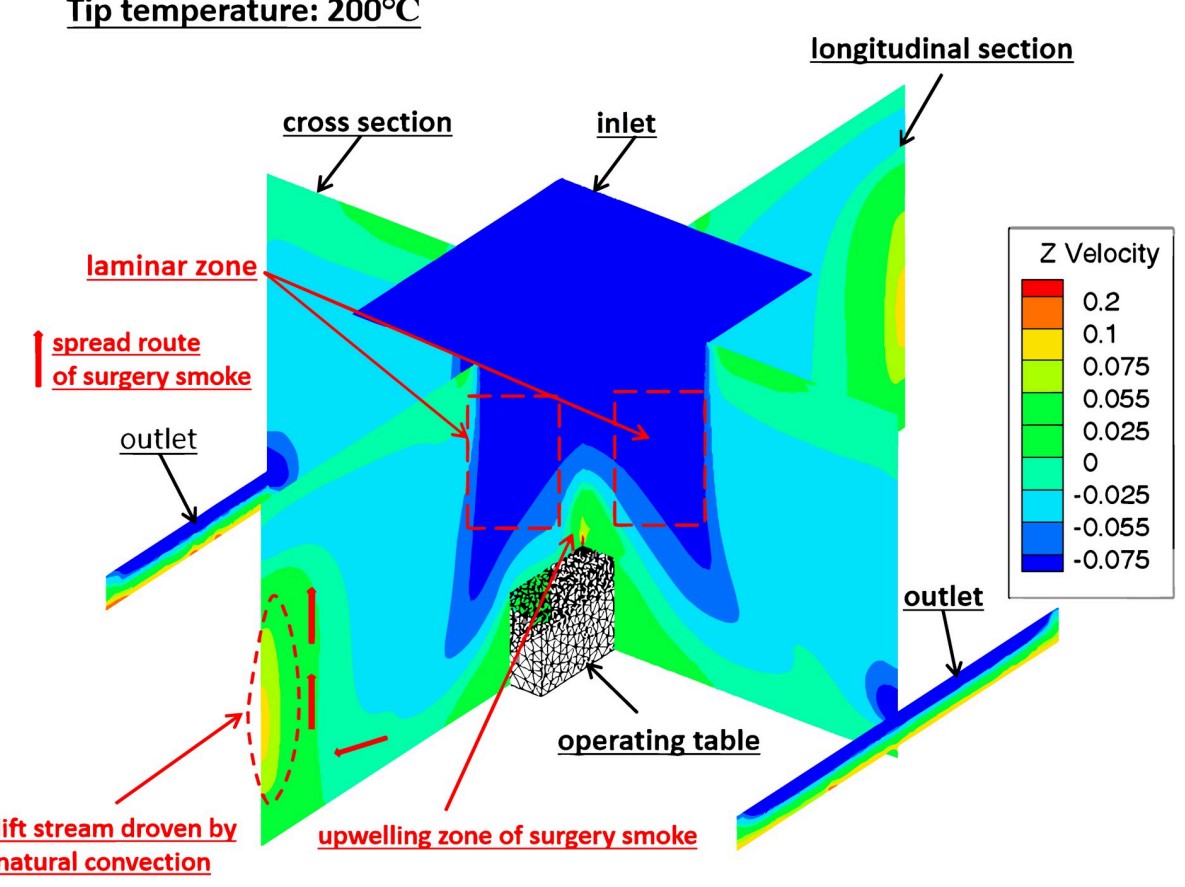

**Fig 8. The contour of velocity Z of gas in theatre during electro-surgery.**

It was shown in Figs 6 and 7 that the formation rate of condensed tar droplets and non-viable particles increased with the rising power of electro-surgical knife. Eqs (15) and (16) denoted the relation between power level of electro-surgical knife and the formation rate of condensed tar droplets and non-viable particles, which were got from polynomial fitting of experiment data.

$$F_{particle} = -4.14 \times 10^{-3} + 6.33 \times 10^{-4}P - 2.06 \times 10^{-6}P^2 + 2.80 \times 10^{-7}P^3 \qquad (15)$$

$$F_{tar} = 2.82 \times 10^{-3} + 4.69 \times 10^{-4}P - 9.69 \times 10^{-6}P^2 + 1.98 \times 10^{-7}P^3 \qquad (16)$$

Where, $F_{particle}$ was the formation rate of non-viable particles during experiment, g/s; $F_{tar}$ was the formation rate of condensed tar droplets during experiment, g/s.

According to Eq (13), when the power of electro-surgical knife was at levels of 20, 60, 70 and 80 W, the knife tip temperature was at levels of 200, 300, 400 and 500°C, respectively. Based on Eqs (14), (15) and (16), the corresponding values of $G_{waste\ gas}$, $F_{particle}$ and $F_{tar}$ to knife tip temperature 200, 300, 400 and 500°C were calculated.

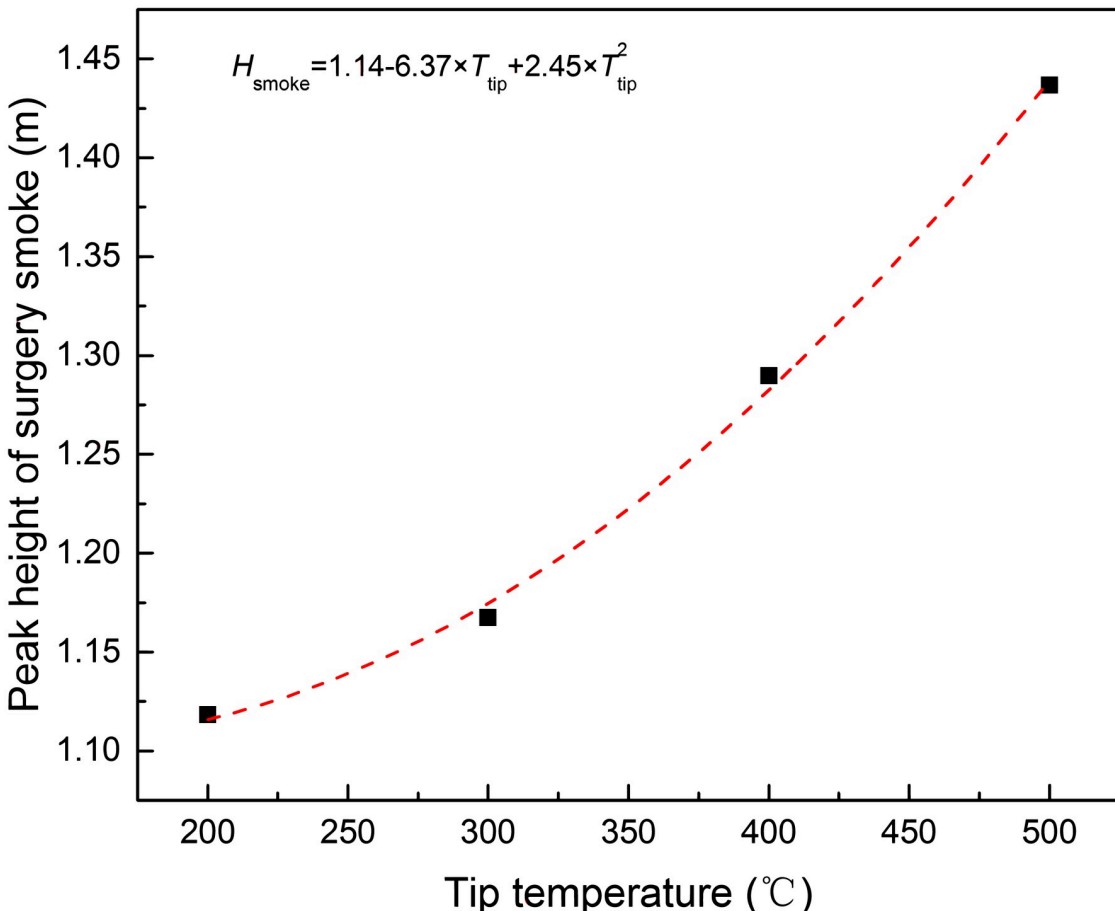

**Fig 9. The peak height of surgical smoke vs tip temperature of electro-surgical knife.**

## Results of simulation of theatre during electrosurgery

Based on the results of in vitro cutting experiment, there were 4 cases of electro-surgery-laparotomy in laminar operation room simulated with FLUENT, and in the cases the tip temperatures of electro-surgical knife were 200, 300, 400, 500˚C.

From the simulation, it was found that the flow field in the theatre during electro-surgery-laparotomy could be divided into 7 zones according to $Z$ velocity (Fig 8): (1) upwelling flow zone right above the incision position of electro-surgical knife, (2) downward laminar zone right above operating table, (4) horizontal flow zone (towards outlets) between operating table and outlets, (5) downward slow flow zone around laminar flow zone, (6) horizontal flow zone (towards walls without outlet) and (7) upward natural convection flow zone nearing the walls without outlet on them. In these zones, the upwelling flow zone right above the incision position was most important, as the surgical smoke sourced in this zone, most contamination was concentrated in this zone and spread to the whole theatre. According to the simulation, the maximum height of upwelling flow zone ($H_{smoke}$) increased as the tip temperature of electro-surgical knife rose, Fig 9. The upward natural convection flow zone nearing the walls without outlet on them was also important, as the surgical smoke spread to this zone, stayed and accumulated in this zone.

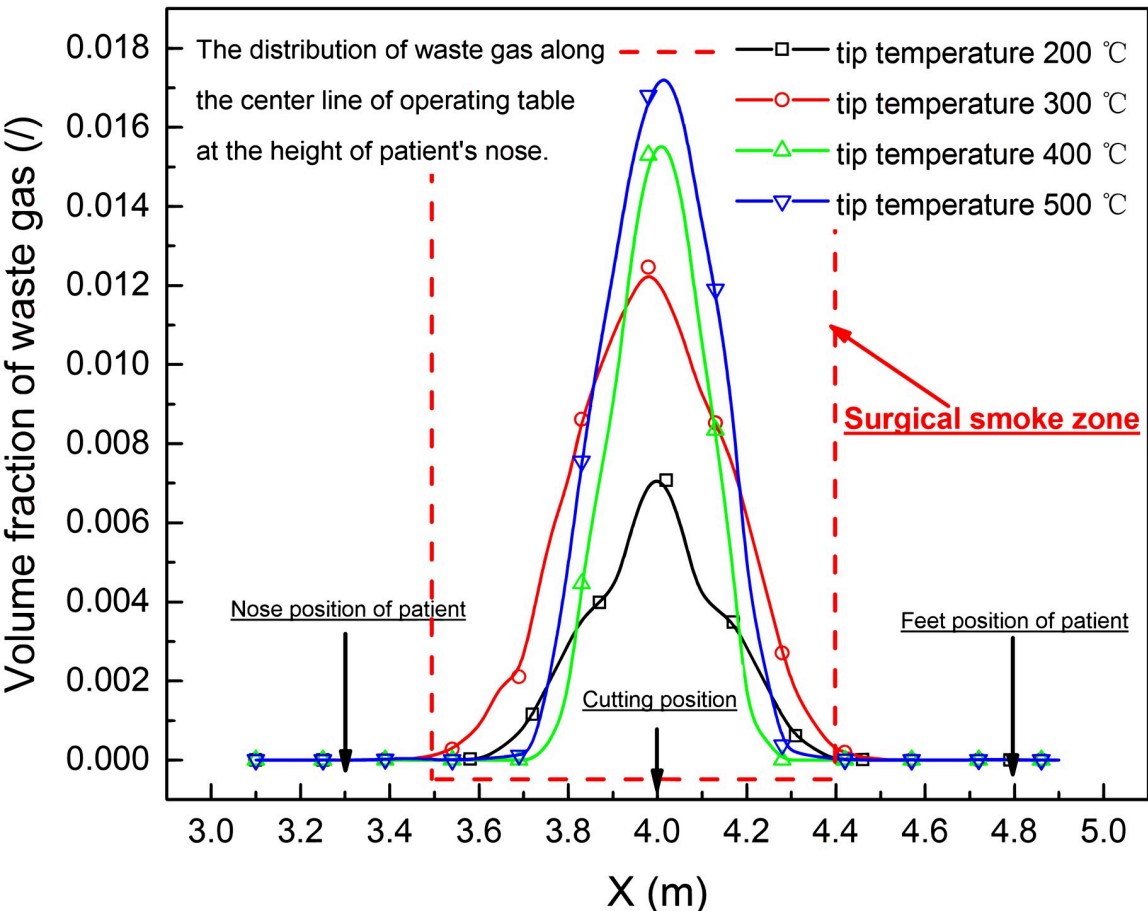

**Fig 10. The waste gas distribution along the center line of operating table at the height of patient's nose ($Z = 1.1$ m).**

The waste gas, non-viable particles and condensed tar droplets distribution along the center line of operating table at the height of patient's nose ($Z = 1.1$ m), who was lying on the operating table, was shown in Figs 10–12. It was found that the concentration of waste gas, non-viable particles and condensed tar droplet was high in the area <0.5 m from the cutting position, and the peak concentration was increased with the rising of tip temperature of electro-surgical unit at height of lying patient's nose.

The non-viable particles and condensed tar droplets distribution along the border of operating table at the height of surgeon's nose ($Z = 1.5$ m), who was standing around the operating table, was shown in Figs 13 and 14. It was found that the concentration of non-viable particles and condensed tar droplet was high near head part and feet part of operating table, and the maximum concentration was increased with the rising of tip temperature of electro-surgical unit at height of standing surgeon's nose around the operating table.

## Discussion

From Figs 3–5, the tip temperature of electro-surgical knife increased with the increasing power of knife, and the rising rate of tip temperature accelerated when power of knife was above 60 W. The relation of formation rate of non-viable particles to power of knife had the

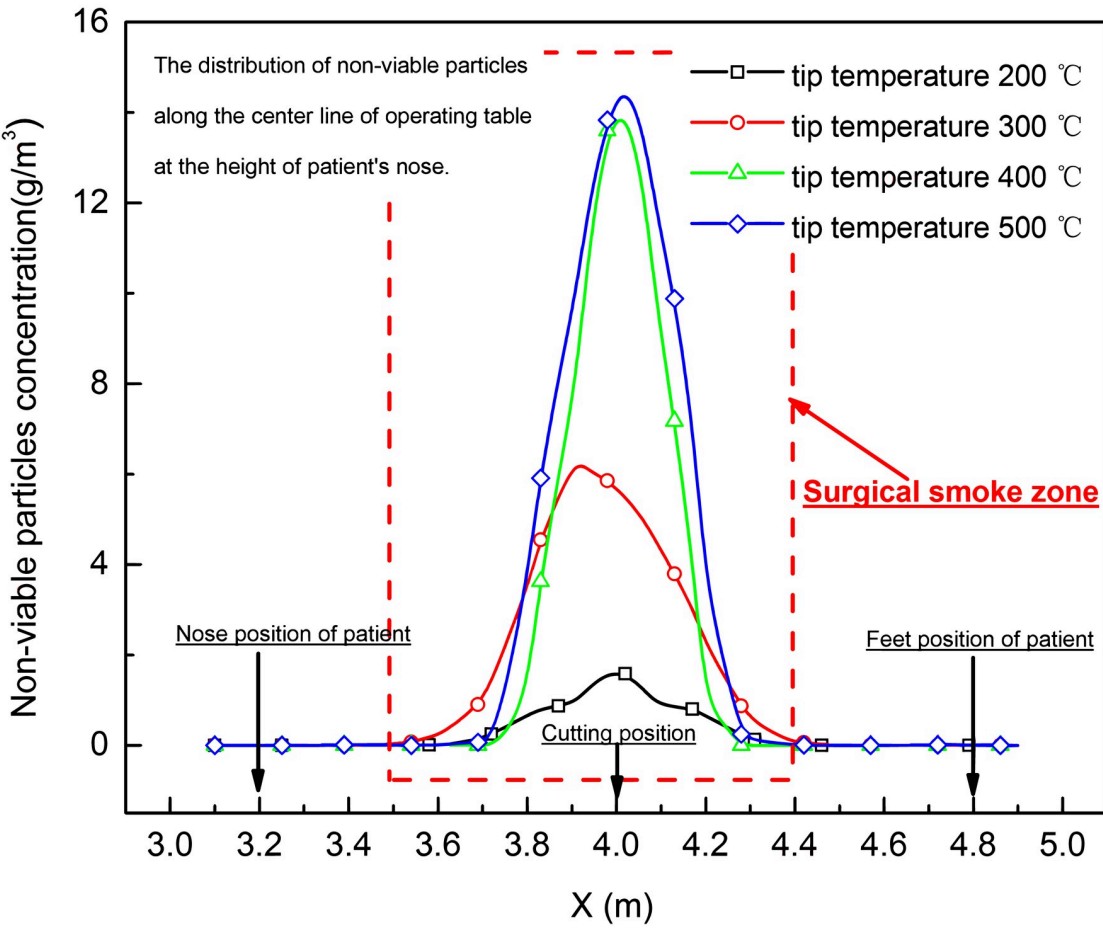

**Fig 11. Non-viable particle distribution along the center line of operating table at the height of patient's nose (Z = 1.1 m).**

similar law. But the formation rate of waste gas and condensed tar droplets increased gradually as power of knife increasing. The main components of waste gas formed from incision of electro-surgical unit were $CO+CO_2$.

Based on the derived flow field of theatre during electro-surgery (laparotomy), Fig 8, it was found: (1) the surgical smoke including waste gas, condensed tar droplets and in-viable particles, upwelled from the incision of electro-surgical knife and the driving forces were upward thrust of new formed waste gas by tissue pyrolysis at high temperature knife tip and buoyancy of hot waste gas; (2) As reversed to downward laminar air flow from the ceiling of theatre, the surgical smoke turned 180 degrees from upward to downward, and the height where velocity $Z$ of gas = 0 was defined as the maximum height of surgical smoke rising right above the incision of electro-surgical unit ($H_{smoke}$); (3) large portion of surgical smoke was then flowed downward to outlets of theatre, and small portion of surgical smoke spread to the head or feet direction of operating table; (4) there was a spread route of dispersed surgical smoke near the walls of theatre without outlet, in which some surgery spread to the walls without outlet, the nature convection near the wall (the wall temperature higher than gas temperature in theatre as heat transfer from outside) drove the dispersed surgical smoke upward into upper part of theatre, and then spread to the whole theatre.

It was also found that the maximum height of surgical smoke rising right above the incision of electro-surgical unit ($H_{smoke}$) was increased with rising tip temperature($T_{tip}$) of electro-

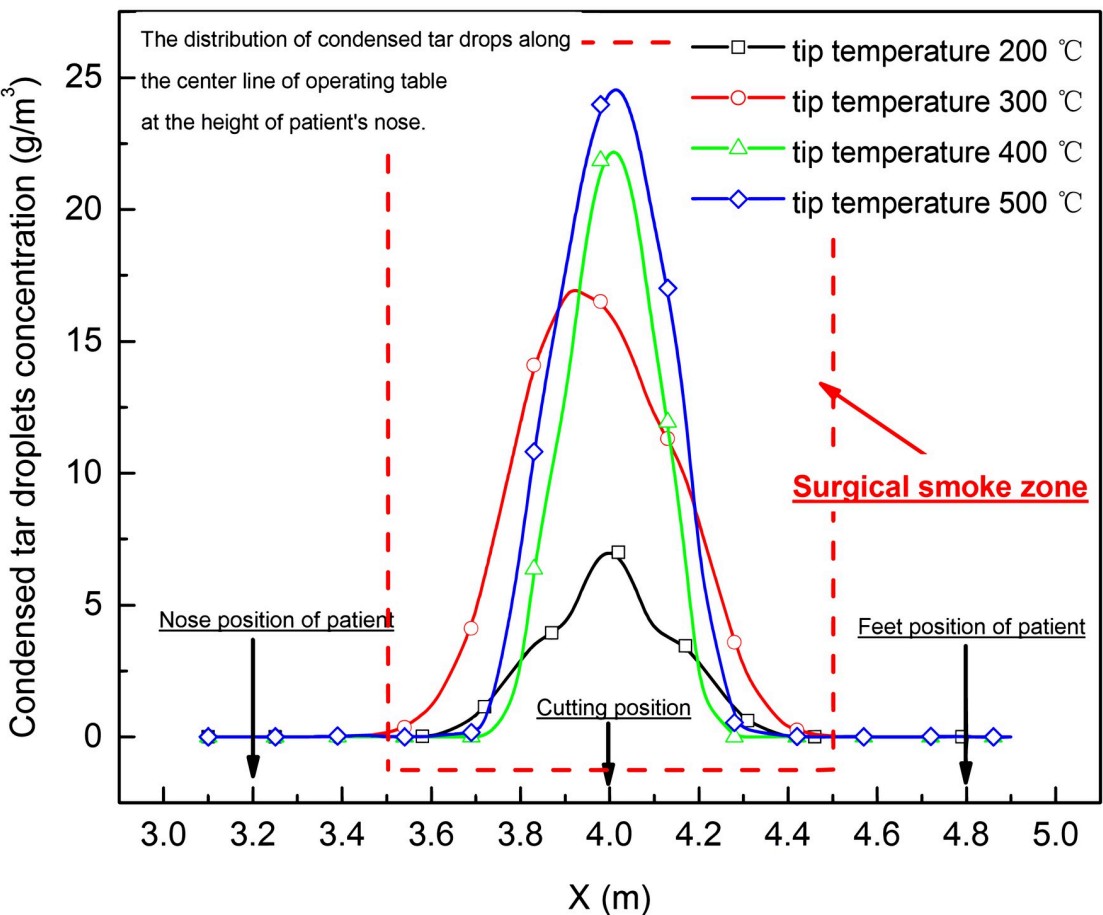

**Fig 12. Condensed tar droplet distribution along the center line of operating table at the height of patient's nose ($Z = 1.1$ m).**

surgical knife, Fig 9, and Eq (17) was the polynomial fit of $H_{\mathrm{smoke}}$ vs $T_{\mathrm{tip}}$ data points.

$$H_{\mathrm{smoke}} = 1.14 - 6.37 \times T_{\mathrm{tip}} + 2.45 \times T_{\mathrm{tip}}^{2} \tag{17}$$

Based on the simulation of electro-surgery (laparotomy) in laminar operation room, the surgical smoke including waste gas, condensed tar droplets and in-viable particles, upwelled from the incision of electro-surgical knife to the height of 1.1–1.45 m, Fig 9, and then turned back to by the falling downward laminar air. In the counter flow area, the waste gas, non-viable particles and condensed tar droplets were concentrated by the upward and downward stream. Within 1.45 m in height and 0.5 m in horizontal from incision position of knife, the concentrations of the waste gas, non-viable particles and condensed tar droplets were high.

During laparotomy, the surgeon should bow into this area, the high contamination of waste gas, condensed tar droplets and in-viable particle should be hazard to health of surgeon. The CO concentrate in the waste gas at the operating zone of surgeon, that is 1.1–1.45 m above the cutting zone, was higher than 50 ppm, when the tip temperature of the electro-surgical unit more than 400˚C, this content is much greater than the 9 ppm upper limit for an 8-hour exposure set by the Unites States Environmental Protection Agency (EPA) [22]. Continuous

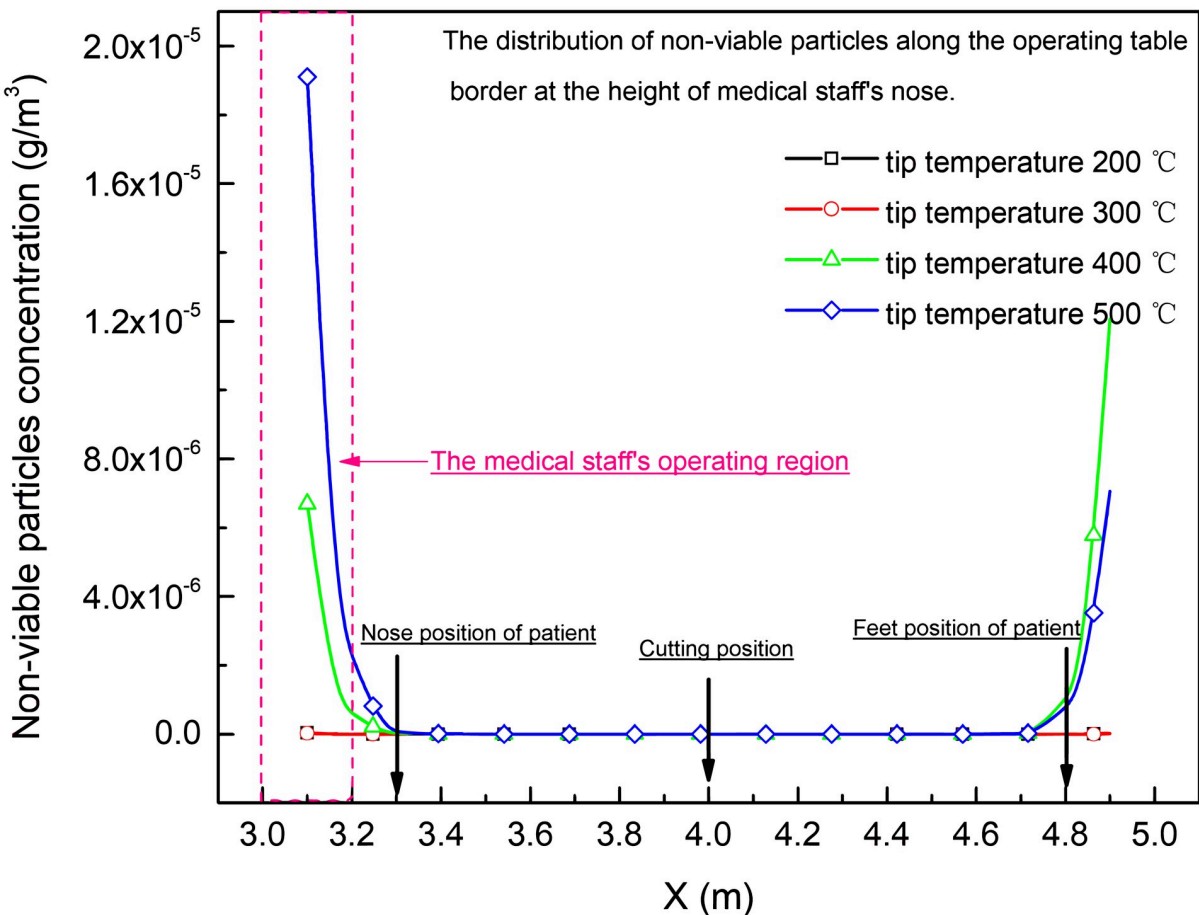

**Fig 13. Non-viable particles distribution along the border of operating table at the height of surgeon's nose (Z = 1.5 m).**

exposure more than 5 days to a this level of CO content leads to a mean the operating surgeon's HbCO level > 2%, which was above the threshold for diagnosis of CO poisoning [23].

As the tip temperature of knife increased from 200 to 500°C, the horizontal range of high contamination area around the incision position of electro-surgical unit was not enlarged, but vertical range of high contamination area around the incision position of electro-surgical unit rose from 1.1 m to 1.45 m, and the concentration of waste gas, condensed tar droplets and non-viable particles increased.

For the patient, the CO concentrate in the waste gas in the wound from the cutting operation, when the tip temperature of the electro-surgical unit more than 400°C, was near 100ppm, this content was also much greater than the 9 ppm upper limit for an 8-hour exposure set by the Unites States Environmental Protection Agency (EPA) [22]. The condensed tar droplets and in-viable particle in the wound from the cutting operation were high than 20 $g/m^3$ and 12 $g/m^3$, which was near that value in the smoke of the cigarette [3]. As the tissue in the wound from the cutting operation is open, the high contamination of waste gas, condensed tar droplets and in-viable particle during electro-surgery should be more hazard to health of patient.

The natural convection near the walls without outlets also influenced the contamination of whole theatre, led waste gas, condensed tar droplets and non-viable particles into the low velocity region outside laminar region under the air inlet. At the nose height of the standing

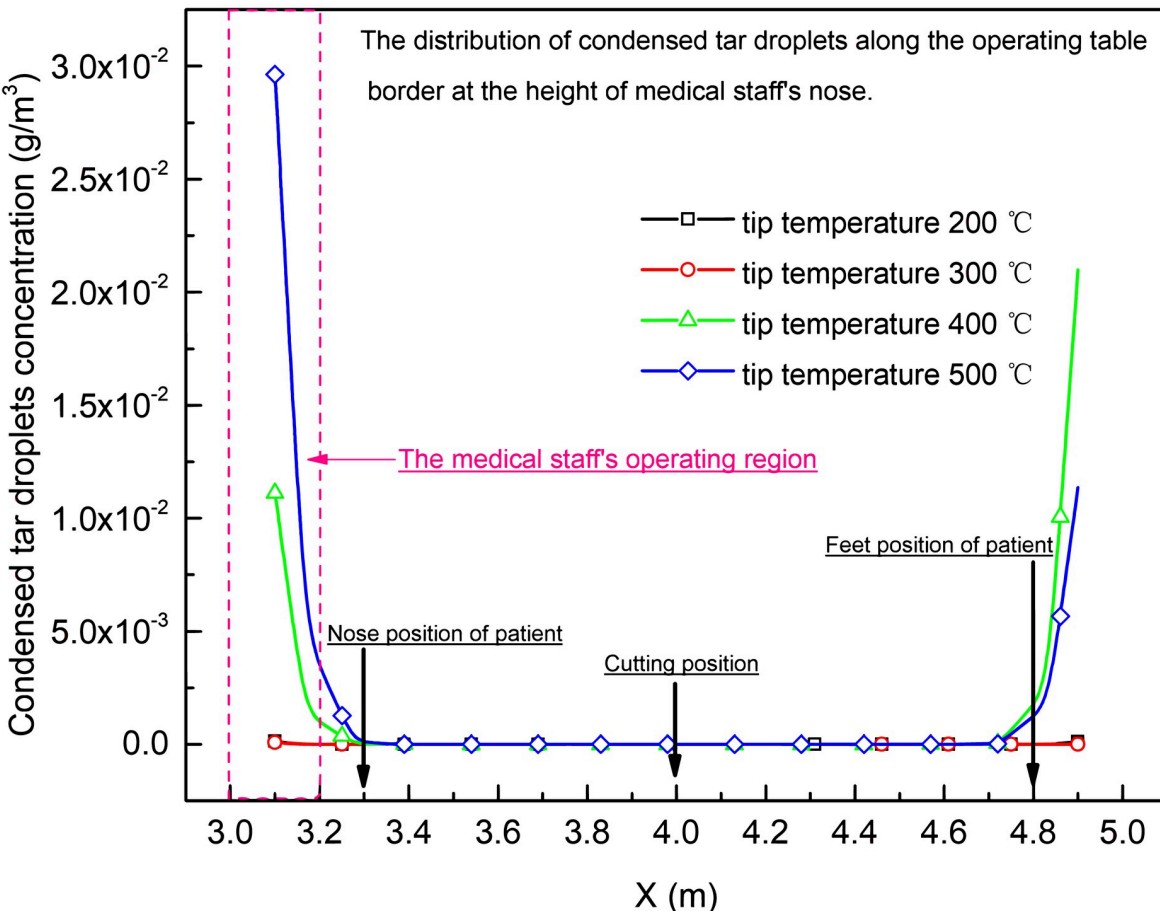

**Fig 14. Condensed tar droplet distribution along the border of operating table at the height of surgeon's nose (Z = 1.5 m).**

surgeon around operating table, the regions near the head or feet of operating table, respectively, were contaminated with higher concentration of non-viable particles and condensed tar droplets. As the tip temperature of electro-surgical knife increased, the concentration of non-viable particles and condensed tar droplets in the regions increased more than 3 times, Figs 13 and 14.

## Conclusion

1. The tip temperature of electro-surgical knife increased with the increasing power of knife, and the rising rate of tip temperature accelerated when power of knife was above 60 W. The polynomial fitting relationship was derived.

2. As the tip temperature of electro-knife increased from 200 to 500˚C, maximum ascending height of surgical smoke right above the incision position of electro-surgical unit increased from 1.1 m to 1.45 m.

3. When the tip temperature of electro-knife was more 400˚C, the CO content in the operating zone of surgeon was more than 200 ppm, which would cause HbCO level increase after multi-day's continuous work.

4. As the tissue in the patient's wound was open during the electro-knife operation with tip temperature more than $400°C$, the content of condensed tar droplets and in-viable particle was higher than $20 \text{ g/m}^3$ and $12 \text{ g/m}^3$, which should be more hazard to health of patient than that of surgeon.

5. The natural convection flow near the walls without outlet caused part of surgical smoke to stay and accumulate in the theatre.

## Supporting information

**S1 Table. Material properties.**
(DOCX)

## Author Contributions

**Conceptualization:** Hui Yu.

**Data curation:** Hui Yu.

**Formal analysis:** Hui Yu.

**Investigation:** Hui Yu.

**Methodology:** Hui Yu.

**Software:** Hui Yu.

**Supervision:** Hui Yu.

**Validation:** Hui Yu.

**Visualization:** Hui Yu.

**Writing – original draft:** Hui Yu.

**Writing – review & editing:** Hui Yu.

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
