## [Decision Letter · Decision Letter 0]

23 Nov 2023

PONE-D-23-32898The research on the effect of temperature of electro-surgical unit to surgical smoke distribution in theatre-in vitro and simulation studyPLOS ONE

Dear Dr. Yu,

Thank you for submitting your manuscript to PLOS ONE. After careful consideration, we feel that it has merit but does not fully meet PLOS ONE’s publication criteria as it currently stands. Therefore, we invite you to submit a revised version of the manuscript that addresses the points raised during the review process.

We look forward to receiving your revised manuscript.

Kind regards,

Jorddy Neves Cruz

Academic Editor

PLOS ONE

Journal Requirements:

Additional Editor Comments:

All modifications made to the manuscript must be marked in yellow.

A point-by-point response must be provided to the reviewer.

Reviewers' comments:

Reviewer's Responses to Questions

**Comments to the Author**

1. Is the manuscript technically sound, and do the data support the conclusions?

Reviewer #1: Yes

Reviewer #2: Yes

2. Has the statistical analysis been performed appropriately and rigorously? 

Reviewer #1: I Don't Know

Reviewer #2: Yes

3. Have the authors made all data underlying the findings in their manuscript fully available?

Reviewer #1: Yes

Reviewer #2: Yes

4. Is the manuscript presented in an intelligible fashion and written in standard English?

Reviewer #1: Yes

Reviewer #2: Yes

5. Review Comments to the Author

Reviewer #1: The abstract should be in a structured format with aims, methodology, results and summary.

The blurred figures should be replaced.

The graphical representations of the results and its discussion should be elaborated in context to clinical applications

Reviewer #2: First of all, I found this study on surgical smoke particle measurement very valuable. I wish you continued work. I have some suggestions other than article modelling.

The size of the pathicle according to the types of cautery should be added here in the literature.

Page 20 line 46: harmful substances: It would be good to explain what these items are.

line 46: not only tissue protein and fat but the vast majority of water, vapour

line 70 : not only surgeons and patients but scrap nurses and anaesthetists. It affects everyone in the room, even if it's not sterile in the surgical field. It should be written with references.

Results: The region of upward natural convection flow with no outlet towards the walls was also important, because it was said that the surgical smoke spread to this region, stayed in this region and accumulated.

There is information in the literature that the particles reached a distance of 1 meter away. this part is also important in terms of the transmission of biological particles, this part should be explained.

Is the chemical content of what is called tar droplets distribution? Is the composition of voc and pah should be written more clearly.

In the result section, it is stated that there is an increase in the number of non-living particles. has this content been fully analysed? because it may also contain biological living materials. hepatitis b transmission is available in the literature. if such a test has not been performed, no definite judgement should be made. if it has been done, it should be explained.

6. PLOS authors have the option to publish the peer review history of their article (what does this mean?). If published, this will include your full peer review and any attached files.

Reviewer #1: **Yes: **Prof. Dr. Karthikeyan Ramalingam

Reviewer #2: No

---

## [Author Response · Author response to Decision Letter 0]

8 Jan 2024

PONE-D-23-32898

In the following content, Q denotes Question from reviewers, A denotes Answer from author.

Reviewer #1: 

(Q1) The abstract should be in a structured format with aims, methodology, results and summary.

(A1) The abstract have been revised to a structured format, as follows.

“Aim

In order to institute effective countermeasures, quantifying of the effect of tip temperature of electro-surgical unit to surgical smoke distribution in theatre was studied. 

Methodology

The relation of tip temperature to power of electro-surgical unit through in vitro cutting experiment. Based on experiment data, the mathematical model was established to simulate the electro-surgery in laminar operating room.

Results

As the tip temperature of knife increased from 200 to 500 ℃, maximum ascending height of surgical smoke right above the incision position of electro-surgical unit increased from 1.1 m to 1.45 m. When the tip temperature of electro-knife was more 400 ℃, the CO content in the medical staff’s operating zone was more than 200 ppm, which would cause the medical staff’s HbCO level increased. As the patient’s tissue in the wound during operation was open, when the electro-knife of more than 400 ℃, the content of condensed tar droplets and in-viable particle was higher than 20 g/m3 and 12 g/m3 in the zone around patient’s wound of open tissue.

Summary

As the power of electro-surgical knife increased, the knife tip temperature increased. Total content of (CO, CO2, CH4, NH3) in waste gas and net flow rate of waste gas at outlet increased with the rising temperature of knife tip and formation rate of condensed tar droplets and non-viable particles also increased. The maximum height of surgical smoke rising right above the incision of electro-surgical unit was increased with rising temperature of electro-surgical knife tip.” 

(Q2) The blurred figures should be replaced.

(A2) Fig2 and Fig8 are changed as follows.

Fig2

Fig8

(Q3) The graphical representations of the results and its discussion should be elaborated in context to clinical applications.

(A3) The representation of the results and discussion have been changed as follow.

Line 256 added: “The chemical analysis of condensed tar droplets was the mixture of benzene, toluene, acrylonitrile, methylpropene and acetaldehyde. The chemical composition of non-viable particles was mostly carbon and a small amount of calcia and magnesia. The sizes of the condensed tar droplets and non-viable particles partly measured in the experiment was between 0.5 µm and 10 µm.”

Line 285 changed: “The upward natural convection flow zone nearing the walls without outlet lead to spreading, staying and accumulating of the surgical smoke to this zone, stayed and accumulated in this zone.”

Line 357 changed: “During laparotomy, the medical staff operated around the operating bed and they need to bow into the high contamination zone of waste gas, condensed tar droplets and in-viable particle should be hazard to health of medical staff.”

Reviewer #2: 

(Q1) First of all, I found this study on surgical smoke particle measurement very valuable. I wish you continued work. I have some suggestions other than article modelling.

(A1) Thank you, Sir. We will carry out more refine research next step.

(Q2) The size of the particle according to the types of cautery should be added here in the literature.

(A2) We have measured some particles collected in the experiments, but not all the particles were measured. So the size distribution of particles was not given in this paper. This sentence “The chemical analysis of condensed tar droplets was the mixture of benzene, toluene, acrylonitrile, methylpropene and acetaldehyde. The chemical composition of non-viable particles was mostly carbon and a small amount of calcia and magnesia. The sizes of the condensed tar droplets and non-viable particles partly measured in the experiment was between 0.5 µm and 10 µm.” was add at Line 256.

(Q3) Page 20 line 46: harmful substances: It would be good to explain what these items are.

(A3) The Line 46 was changed to “Many studies show that it contained vapour, harmful substances, e.g. benzene, toluene, carbon monoxide and lung-damaging particulates [1-4].” 

(Q4) line 46: not only tissue protein and fat but the vast majority of water, vapour

(A4) line46 was changed as “Surgical smoke was sourced from ablation of tissue water, protein and fat.”.

(Q5) line 70 : not only surgeons and patients but scrap nurses and anaesthetists. It affects everyone in the room, even if it's not sterile in the surgical field. It should be written with references.

(A5) In this paper the word “surgeon” have been changed as “Medical staff” in all sections and figures, which includes surgeons, nurses and anaethetists etc in the operating room. And reference 22 was added to the paper: “22. Ball K, Gilder R. A Mixed Method Survey on the Impact of Exposure to Surgical Smoke on Perioperative Nurses[J]. Perioperative Care and Operating Room Management. 2021; 26(3): 1-9. doi:10.1016/j.pcorm.2021.100232.”.

(Q6) Results: The region of upward natural convection flow with no outlet towards the walls was also important, because it was said that the surgical smoke spread to this region, stayed in this region and accumulated.

(A6) According to the model research in this paper, the accumulation of surgical smoke near the wall without outlets was discussed. And Line 380 was changed as “The natural convection near the walls without outlets led to surgical smoke accumulated in the region 1 m from the bed, for non-viable particles increased to 1.9×10-5 g/m3 and the condensed tar droplets 0.03 g/m3, this also influenced the contamination of whole theatre, led waste gas, condensed tar droplets and non-viable particles into the low velocity region outside laminar region under the air inlet. At the nose height of the standing medical staff around operating table, the regions near the head or feet of operating table, respectively, were contaminated with higher concentration of non-viable particles and condensed tar droplets.”.

(Q7) There is information in the literature that the particles reached a distance of 1 meter away. this part is also important in terms of the transmission of biological particles, this part should be explained.

(A7) As the model research in this paper, the region 1 meter at the direction to the wall without outlet from the operating bed was the surgical smoke accumulating region and medical staff operating region, as new Fig13 and Fig14. But in our research the biological particles were not measured and this part was not discussed. 

(Q8) Is the chemical content of what is called tar droplets distribution? Is the composition of voc and pah should be written more clearly.

(A8) The content of tar droplets was partly measured and Line 256 was added: “The chemical analysis of condensed tar droplets was the mixture of benzene, toluene, acrylonitrile, methylpropene and acetaldehyde. The chemical composition of non-viable particles was mostly carbon and a small amount of calcia and magnesia. The sizes of the condensed tar droplets and non-viable particles partly measured in the experiment was between 0.5 µm and 10 µm.” 

(Q9) In the result section, it is stated that there is an increase in the number of non-living particles. has this content been fully analysed? because it may also contain biological living materials. hepatitis b transmission is available in the literature. if such a test has not been performed, no definite judgement should be made. if it has been done, it should be explained.

(A9) The size of non-viable particles was partly measured, its weight was measured, however its size distribution was not discussed. We will research this content in the future. The chemical analysis of the non-viable particles was measured as the mixture of mostly Carbon and small amount of Oxides, however the biological living materials on the particles was not measured. Thanks for the suggestions from the reviewers, we will research in this content. Thanks again.

---

## [Editor Report · Decision Letter 1]

9 Feb 2024

The research on the effect of temperature of electro-surgical unit to surgical smoke distribution in theatre-in vitro and simulation study

PONE-D-23-32898R1

Dear Dr. Yu,

We’re pleased to inform you that your manuscript has been judged scientifically suitable for publication and will be formally accepted for publication once it meets all outstanding technical requirements.

Kind regards,

Jorddy Neves Cruz

Academic Editor

PLOS ONE
---

## [Editor Report · Acceptance letter]

23 Feb 2024

PONE-D-23-32898R1 

PLOS ONE

Dear Dr. Yu, 

I'm pleased to inform you that your manuscript has been deemed suitable for publication in PLOS ONE. Congratulations! Your manuscript is now being handed over to our production team.

Kind regards, 

on behalf of

Dr. Jorddy Neves Cruz 

Academic Editor

PLOS ONE